# A Review of Chemical Contaminants in Marine and Fresh Water Fish in Nigeria

**DOI:** 10.3390/foods10092013

**Published:** 2021-08-27

**Authors:** Abimbola Uzomah, Anne-Katrine Lundebye, Marian Kjellevold, Fubara A. Chuku, Oluwafemi A. Stephen

**Affiliations:** 1Department of Food Science and Technology, Federal University of Technology, Owerri, P.M.B. 1526, Owerri 460001, Nigeria; 2Institute of Marine Research, P.O. Box 2029 Nordnes, 5817 Bergen, Norway; anne-katrine.lundebye@hi.no; 3Food Safety and Quality Programme, Federal Ministry of Health, Abuja, P.M.B. 083, Abuja 900104, Nigeria; fubarachuku@yahoo.com (F.A.C.); femistephen@live.co.uk (O.A.S.)

**Keywords:** Nigeria, PAHs, POPs, metals, microplastics, marine fish, freshwater fish, pollutants, contaminants

## Abstract

Pollutants in aquatic food are a major global concern for food safety and are a challenge to both national and international regulatory bodies. In the present work, we have reviewed available data on the concentrations of polycyclic aromatic hydrocarbons (PAH), persistent organic pollutants, metals, and microplastics in freshwater and marine fish in Nigeria with reference to international maximum levels for contaminants in food and the potential risk to human health. While most of the contaminant levels reported for fish do not imply any health issues, iron and lead may represent potentially toxic levels in fish from specific areas. Studies on PAHs in marine fish are scarce in Nigeria, and the main focus is on the environmental pollution caused by PAHs rather than on their presence in food. The findings suggest that the consumption of smoked *Ethmalosa fimbriata* poses a higher potential carcinogenic risk than the other fish species that were investigated. Most of the other studies on PAHs in smoked fish are focused on the smoking method, and little information is available on the initial level of PAHs prior to the smoking process. Metal contamination in fish appeared to be affected by mineral deposits in the environment and industrial effluents. In general, heavy metal levels in fish are below the maximum levels, while there is limited data available on POPs of relevance to food safety in fish from Nigeria, particularly in terms of dioxins, brominated flame retardants, and fluorinated compounds. Furthermore, there is currently limited information on the levels of microplastics in fish from Nigerian waters. This work revealed the need for a more systematic sampling strategy for fish in order to identify the most vulnerable species, the hot spots of contaminants, and applicable food safety control measures for fish produced and consumed in Nigeria.

## 1. Introduction

Nigeria is a vast area of land in West Africa bordered by the Gulf of Guinea and the Bight of Benin along the coastal region. There is an active fishing industry along the coast, which covers the offshore waters between the 30-mile limit and the territorial 200-mile exclusive economic zone (EEZ).

The coastal area is the hub of marine fish production, and the fishing business constitutes a large source of income for many and is the major source of dietary protein for the rural population [1]

### Fishery Practices in Nigeria

The coastal region has more than 104 different species of fish of commercial importance (Table 1). Furthermore, inland freshwaters have about 196 species of fish belonging to 105 genera and representing 46 families. These species are dominated by *Pseudotolithus* spp. (croakers), *Brachydeuterus* spp. (grunts), various *Cynoglossus* spp. (sole), and Arius spp. (catfish) [2,3,4,5].

Fish production comes from three main sources, namely artisanal, industrial (trawlers), and domestic (aquaculture) fisheries. Artisanal fisheries are well-established along the coastal areas, creeks, lagoons, inshore, and the inland waterways. They are a better income-generating source for the communities along the coastal area than the industrial fisheries [6,7,8]. The fish production in Nigeria from 2010 to 2015 is summarized in Table 2.

Industrial fish production is the lowest, while artisanal fish capture covers the coastal and brackish water as well as the rivers and lakes and is much larger than the aquaculture and industrial sectors (Figure 1) [8]. The torpedo-shaped catfish, *Clarias* spp. is the most produced fish. Other farmed species with high production are the *Hemichromes/Oreochromis*, *Heterbranchus*, *Cyprinidae*, and *Osteichthye* species. The production volumes reflect the consumption patterns of the local people [1,7,9,10,11,12,13]. In 2013, it was reported that fish made up 40% of the country’s protein intake, with an estimated consumption of 13.3 kg/person/year [11]. There is, however, limited information on the accuracy of consumption data. The study by Adeniyi et al. (2012) [12] on the consumption patterns of fish among households was for a small population in a local government area in Ibadan, Oyo State. These authors associated the low socio-economic status of the people with their choice of fish, which was found to be the most affordable animal protein in the study area. A study of a larger sample size with wider geographical coverage is required for a better understanding of the fish consumption patterns of Nigerians.

Over the few past years, there has been a gradual decline in the populations of the fish species listed in Table 1 [4]. This has been attributed to overfishing and environmental pollution [5,13,14]. Some of the pollution sources are from anthropogenic activities such as oil exploration, which include pipeline rupture, oil well blowouts, seepages, tanker accidents, ballast water and refinery wastewater, sabotage of operational facilities, oil spillage, and gas flaring [15,16]. These pollutants include PAHs, persistent organic pollutants (POPs), pesticides, metals, and, more recently, plastic waste.

This report reviewed available data on contaminants in freshwater and marine fish in Nigeria. It reflects the degree of the environmental pollution in fish consumed in Nigeria with reference to the international maximum levels and the potential risk to human health. The report is divided into four parts based on the nature of the contaminants:(i)Poly Aromatic Hydrocarbons (PAHs);(ii)Persistent Organic Pollutants (POPs);(iii)Metals;(iv)Microplastics.

## 2. Materials and Methods

Relevant peer-reviewed publications were sourced for information regarding the contaminants in marine and freshwater fish consumed in Nigeria. Academic search engines that were accessed include, Proquest, ScienceDirect, Google Scholar, Microsoft Academy, and CORE, inter alia. Keywords including Nigeria, PAHs, POPs, metals, microplastics, marine-fish, freshwater fish, pollutants, and contaminants were used. The most relevant publications were those with analytical data on specific contaminants in fresh and marine water fish consumed by Nigerians. Similar data within the same period (year and season) or same location were collectively reported. Selection was also based on the year of publication, and recent publications were prioritised, except where the chronological information is important. Over 300 articles were accessed, and 130 were found to contain relevant information. A summary of the fish species analysed for contaminants is presented in Table 3. The analytical method used to determine the PAHs and POPs was gas chromatography mass spectrometry (GC/MS). The different studies applied different detectors such as flame ionization detectors (FID), electron capture detectors (ECD), and mass selective detectors (MSD). HPLC was rarely used. Metal analysis was performed using atomic absorption spectroscopy (AAS) and inductively coupled mass spectrometry (ICP-MS). Studies on the detection of microplastics in fish tissue were limited, and the analytical procedure used a fluorescence stereo zoom microscope (FSZM) for the analyses. The sampling schemes, handling, and storage were conducted according to standard procedures (Table 3).

## 3. Results and Discussion

### 3.1. Polycyclic Aromatic Hydrocarbons (PAHs)

The major sources of environmental PAH contamination have been largely attributed to anthropogenic activities, particularly in the oil producing states in Nigeria [50,51].

PAHs and their derivatives can be characterized by their genotoxic and carcinogenic potential. The 16 PAHs that are usually analyzed for environmental pollution are naphthalene (Nap), acenaphthylene (Acy), acenaphthene (Ace), fluorene (Fle), phenanthrene (Ph), anthracene (An), fluoranthene (Fla), pyrene (Py), benzo[a]anthracene (BaA), chrysene (Chr), benzo[b]fluoranthene (BbF), benzo[k]fluoranthene (BkF), benzo[a]pyrene (BaP), indeno [1,2,3-cd]pyrene (InD), dibenz[a,h]anthracene (DahA), and benzo[ghi]perylene (BghiP) [20]. These are all exogenous PAHs from polluted agricultural land and air (atmospheric). However for food (and human health), the European Food Safety Authority (EFSA) included the following PAHs based on concerns for human health: benz[a]anthracene, benzo[a]pyrene, benzo[b]fluoranthene, benzo[j]fluoranthene, benzo[k]fluoranthene, benzo[ghi]perylene, chrysene, cyclopenta(c,d)pyrene, dibenz[a,h]anthracene, dibenzo(a,e)pyrene and dibenzo(a,h)pyrene, dibenzo(a,i)pyrene, dibenzo(a,l)pyrene, indeno(1,2,3,-cd)pyrene, 5-methylchrysene, and benzo(c)fluorene [52].

#### 3.1.1. PAHs in Fish from Fresh and Brackish Water

The degree of contamination is strongly associated with the level of pollutants in the water; hence, many studies have focused on monitoring the levels of PAHs in different aquatic environments. There are three rivers (Sime, Kporghor and Iko) in Akwa-Ibom state in the Niger Delta region of Nigeria that are well known for petroleum pollution from refinery and pipeline vandalization [24]. From these locations, 16 PAHs (naphthalene, acenaphthylene, acenaphthene, fluorene, phenanthrene, anthracene, fluoranthene, pyrene, benzo[a]anthracene, chrysene, benzo[b]fluoranthene, benzo[k]fluoranthene, benzo[a]pyrene, indeno[1,2,3-cd]pyrene, dibenzo[a,h]anthracene, and benzo[g,h,i]perylene) were analyzed in the edible tissues of 30 species of fish and other seafood (periwinkles (*Littorina littorea*) and oysters (*Crassostrea virginica*)) commonly consumed in these communities [24]. The mean PAH concentration in *P. koelreuteri* from the Iko coastal waters was 49 µg/kg (wet weight). This value exceeded the EU maximum level for sum PAH4 (sum of (BaP, BaA, BbF, and chrysene) in smoked fish of 12 µg/kg wet wt. [53]. From the Sime River, the detected PAHs ranged from below the level of detection (LOD) to 22 µg/kg wet wt. in *Littorina littorea*, from the LOD to 87 µg/kg wet wt. in *Crassostrea virginica*, and from LOD to 171 µg kg^−1^ wet wt. in *Periophthalmus koelreuteri*. The highest average concentration of 171 g kg^−1^ wet wt. was recorded for Indeno from Sime waters. These rivers are generally heavily polluted by anthropogenic activities in the surrounding communities, which has resulted in the different levels of PAHs in the fish samples. In the Sime River, *C. virginica* accumulated significantly lower (*p* < 0.05) concentrations of total PAHs than *L. littorea* and *P. koeleuteri* [18].

There were twelve fish samples from the brackish water of Lagos Lagoon (a highly polluted site) that had high levels of PAHs (di, tri and tetra-aromatic isomers of naphthalene, acenaphthylene, phenanthrene, anthracene, fluoranthene, pyrene, benzo(a)-anthracene, and chrysene) [19]. The fish species investigated were *Caranx hippos*, *Mugil cephalus*, *Sphyraena barracuda*, *Sarotherodon melanotheron*, *Tilapia guineensis*, *Ethmalosa fimbriata*, *Tarpon atlanticus*, *Scomberomorus trito*, *Lutjanus agennes*, *Pomadasys jubelini*, *Chrysichthys nigrodigitatus*, *and Lutjanus dentatus.* The highest mean concentration of PAHs (275 µg/kg dry wt.) was found in *Mugil cephalus* while 48 and 30 ng/g dry wt. were reported for *Chrysichthys nigrodigitatus* and *Tilapia guineensis*, respectively.

Studies on PAHs in marine fish are scarce in Nigeria, as the main focus is on environmental PAH pollution rather than on their presence in food. The detrimental effects of these contaminants on fish populations have been reported [17,19]. However, the effects of different remediation actions on PAH levels in fish have not been fully documented.

#### 3.1.2. PAHs in Smoked Fish

PAHs can also be produced during food processing such as grilling, roasting, smoking, and barbecuing [21,22,34]. Fish smoking is highly practised as a means of prolonging shelf-life, enhancing flavour, and increasing utilization [26]. Nigeria produces 194,000 metric tons of dried fish annually, 61% of which is smoked fish [21]. Since smoking is a major source of PAH contamination in fish, the health risks associated with the consumption of smoked fish in Nigeria may be high [30,32]. The impact of the smoking techniques on the amount and type of PAHs that are generated, however, varies [24]. Silva et al. [21], reported the effect of using sawdust, charcoal, and firewood for smoking on the PAH levels in three species of fish (*Arius heude loti*, *Cynoglossus senegalensis* and hake). While charcoal with minimal production of smoke, gave the lowest concentration of sum PAHs, the sawdust on the other hand gave the highest level of PAHs. This was attributed to the pyrolysis of cellulose, hemicellulose and lignin and the limited availability of oxygen at the high processing temperature (>7000 C) [34,36]. The sum PAHs obtained is also related to the oil content of the fish species [26,38]. During fish smoking, the fish oil drips into the fire, and pyrolytic compounds are released. However, this can be controlled if the oil is prevented from dripping into the fire during the smoking process.

A similar study using traditional smoking methods, was carried out by Ubwa et al., (2015) [23]. Five fish species *Arius heudeloti*, *Cynoglussuss enegalensis*, *Clarias gariepinus*, *Blunt hwake* and *Mud minnow* were obtained from a fish farm in Benue state and analysed for the presence of 16 priority PAHs [23]. The results showed the highest sum PAH concentration in fish smoked with sawdust. The PAH levels in fish smoked with sawdust ranged from 815–1550 μg/kg, followed by fish smoked with firewood (738–994 μg/kg) and charcoal producing fish with the lowest PAH concentrations (135–614 μg/kg). The benzo(a)pyrene (BaP) concentration in *Arius heude loti* was 5.7 μg/kg and the BaP concentration in mud minnow was 5.4 μg/kg, a direct effect of using sawdust [23]. These values exceed the EU maximum level for BaP in smoked fish of 2 μg/kg [53]. In line with these findings, Tongo et al., (2017) [25] reported the presence of PAHs in the tissue of four smoked fish species (*Clarias gariepinus*, *Ethmalosa fimbriata*, *Tilapia zilli*, and *Scomber scombrus*) obtained from three major markets (Oreogbe, New Benin and Santana markets) in Edo State. *S. scombrus* had the highest sum PAHs concentration while the sum PAHs concentrations were 0.7, 1.0, 0.7, and 3.6 μg/kg in *C. gariepinus*, *T. zilli*, *E. fimbriata*, and *S. scombrus*, respectively. The estimated cumulative excess cancer risk index for *E. fimbrata* and *C. gariepinus* were higher than that of the other smoked fish species and the values exceeded the USEPA’s acceptable cancer risk level of 10^−6^ [25,41]. The findings suggest that consumption of smoked *E. fimbriata* poses a higher potential carcinogenic risk than the other fish species investigated. Most of the studies on PAHs in smoked fish are focused on the smoking method and little information is available on the initial level of PAHs prior to the smoking. In some other cases, the source of the smoked fish and the method of smoking are not indicated [25].

### 3.2. Persistent Organic Pollutants (POPs)

The environmental pollutants commonly called Persistent Organic Pollutants (POPs) include compounds previously synthesized for use as pesticides and halogenated industrial compounds. These compounds can resist chemical and microbial degradation and can persist for a long time in the environment. Being lipophilic in nature, they readily bioaccumulate in the fatty tissues of organisms except for perflourinated contaminants that bind to proteins. These compounds can be transported far beyond the point of use or application.

POPs can be categorized into four major groups [43]:Pesticides (organochlorine pesticides, OCPs): aldrin, chlordane, DDT, dieldrin, endrin, heptachlor, hexachlorobenzene, mirex, toxaphene;Industrial chemicals: hexachlorobenzene (HCB), polychlorinated biphenyls (PCBs);By-products: hexachlorobenzene, polychlorinated dibenzo-p-dioxins (PCDD) and polychlorinated dibenzofurans (PCDF), and dl-PCBs;Brominated flame retardants and perfluorinated compounds.

Organochlorine pesticides (OCPs) have been widely used by farmers for pest control. They are synthetic, non-polar, toxic, and environmentally persistent dichlorodiphenylethanes, cycodienes or chlorinated benzenes. The PCDD, PCDF, and PCBs, are a family of complex chlorinated compounds with similar structures and biological activity of which 29 (7 of the 75 PCDD compounds, 10 of the 135 PCDF compounds, and 12 of the 209 PCB compounds) have been identified as having dioxin-like toxicity [45]. The health challenges associated with these chemicals led to the ban of PCB [47], however dioxins are unintentional by-products. Brominated flame retardants, mainly the polybrominated diphenyl ethers (PBDE (e.g., pentabromodiphenyl ether (PentaBDE), and octabromodiphenyl ethers (OctaBDE)) belong to a class of POPs banned in the Stockholm Convention [49]. These were subsequently replaced with novel brominated flame retardants (NBFRs) and organophophorus flame retardants (OPFRs) [51].

Most POPs (with the exception of those which are by-products, e.g., dioxins) are components of different industrial and commercial appliances as well as additives in pesticides, plasticizers in paints, plastics, rubber products, etc. [54]. The challenge in Nigeria, and globally, is the resistance of these compounds to degradation and the leaching from disposal systems and landfills [55,56]. In Nigeria, there are efforts both from national regulatory agencies and international intervention to deal with POPs in landfills [57]. Humans are exposed to these compounds through the food chain, inhalation of air (outdoors, indoors and at the workplace), and from occupational and accidental exposure [27]. In water, they are taken up by phytoplankton, then fish (bioaccumulation and bio-magnification processes particularly in oily fish). The concentrations of POPs in fish differs among ecosystem (marine versus inland waters) and may be lower in fish from aquaculture because of the controlled environment and feed composition.

#### 3.2.1. POPs in Fish from Different Water Bodies

##### Fresh Water Sources

Ibor et al., 2016 [58], analyzed PCB concentrations in muscle tissue of tilapia species (*Tilapia guineensis*, *Sarotherodon galileaus* and *Oreochromis niloticus*) from four sites along the Ogun River. Significantly higher PCB concentrations were found in fish from the three polluted sites compared to the control site, with sum PCB (15 congeners) varying between 359–4636 μg/kg and 24–28 μg/kg, respectively. A causal relationship between endocrine disruption and contaminant burden (PCB and lindane) was observed in the tilapia species from the Ogun River [59]. The source of the contaminants was attributed to industrial activities and runoff to the river from the surrounding farmlands [60].

There were two fish species (*Tilapia zilli and Clarias gariepinus*) from the Illushi, Owan and Ogbesse rivers in Edo State that were analysed for pesticide residues: α-BHC, γ-BHC (lindane), β-BHC, heptachlor, aldrin, heptachlor epoxide, endosulfan, dieldrin, endrin, DDT, atrazine, phosphomethlglycine, and canbofuran [29]. The study lasted for 18 months and aimed to assess the possible effects of seasonal variation. The levels of pesticide residues were higher in the tissues of *Clarias gariepinus* (5.5–10 μg/g wet weight) than in *Tilapia zilli* (3.5–5 μg/g wet wt.), while the most dominant residue was the persistent organochlorine ΣBHC. These authors attributed the occurrence and concentration of the residues in the fish samples to the feeding mode, age, and mobility of the biota [61,62]. *Clarias gariepinus* was mainly exposed to the contaminants from foraging in the sediments [63]. In a similar study, Chukwuka et al., (2019) [64], assessed pesticide levels and pathological alterations in reproductive tissue in three fish species (one pelagic: *Tilapia zilli* (*n* = 92) and two benthic: *Neochanna diversus* (*n* = 59), *Clarias gariepinus* (*n* = 68)) from the Owan River in Edo State, which receives run-off from surrounding farmland treated with pesticides. Their findings showed that pesticide levels were higher in the tissues of benthic species than in pelagic fish. They also observed damage and anomalies in ovarian and testicular tissues in both the benthic and pelagic fish samples and attributed this to exposure to pesticides in the surface water and sediment [63]. The levels of the pesticide residue in the water exceeded the maximum limits (EPA, 2004), as cited by Adeboyejo 2011 [63]. Moslen et al., (2019) [28], reported a strong correlation between pesticide (dieldrin and endrin) concentrations in fish samples and sediment.

Endocrine disruptive contaminants and alterations in reproductive development were assessed in the tilapia *Sarotherodon melanotheron* [33]. Sediment samples and a total of 155 fish (74 males and 81 females) were collected from selected sites along the Lagos Lagoon, where two of the locations were heavily polluted, and the third site had some degree of pollution control. The results showed significantly higher concentrations of lindane, dieldrin, 4-iso-nonylphenol, 4-t-octylphenol, and monobutyltin the two polluted sites. The authors correlated the endocrine responses in fish to the contaminant concentrations in the sediment, and these reports corroborated the findings of Ibor et al. (2017) [59]

The occurrence and effect of persistent organochlorine pesticides in the ecosystem is of major concern, not only in Nigeria, but also in several other African countries [65].

##### Brackish Water

In 2008, Adeyemi et al. [35] found concentrations of OCPs in *Tilapia zilli* (red belly tilapia), *Ethmalosa fimbriata* (bonga shad), and *Chrysichthys nigrodigitatus* (catfish) from the Lagos Lagoon and delta to be below the residue limit of 5 mg/kg set by CODEX, (1997), except in the case of HCHs. Similar findings were reported by Adeyemi et al. (2009) [31], whereas higher concentrations of organochlorine pesticides were found in croaker fish, *Pseudotolithus semegalensis*, and *Pseudotolithus typus* from the Lagos Lagoon, [31,35]. The dominant BHC was β-BHC, in the order of β-BHC > lindane > δ-BHC > α-BHC. The total DDT concentration followed the order p,p′-DDT > p,p′-DDD > p,p′-DDE. The high p,p′-DDT levels detected in this study were in contrast to earlier findings [66], which showed that p,p′-DDE was the major DDT metabolite in aquatic species. The concentrations of these OCPs were however found to be below the maximum limits of the FAO/WHO (2005) and USEPA (2006) [67,68].

Pesticide residue in fish from brackish water was found to be higher in the dry season than in the wet season [69], and a similar observation has been reported for fish from pond water [70]. Williams and Unyimadu (2013) [71] related the bioaccumulation of OCPs to the type of fish species and gender variation of the fish in brackish water. They found higher concentrations of OCPs in African Moony (*Psettias sebae*) than in Bonga fish (*Ethmalosa fimbriata*) and higher concentrations in males than in females. The sum concentration of organochlorine pesticides in female *Ethmalosa fimbriata* was 5.7 μg/kg (wet wt.) and the maximum found in male fish was 3005 μg/kg.

Williams and Anake (2013) [72] observed higher concentrations of organochlorine pesticides in snapper (*Lutjanus goreensis*), herring (*Sardinella maderensis*), and oarfish (*Regalecus glesne*) from the brackish water of the Lagos Lagoon during the dry season. The order of the pesticide concentrations in muscle tissues of these species was *Regalecus glesne > Sardinella maderensis > Lutjanus goreensis*. The highest chlorinated hydrocarbon concentration (6181 μg/kg wet wt.) was in *Regalecus glesne*. This concentration was higher than the levels recorded by other researchers [39,70,71,73] in fish samples from various sites along the coastal region from Lagos and Ogun States in the South-West [67] and along the coastal region of the South-South and South-East region of Nigeria [68,69,71].

Unyimadu et al. (2018) [39] investigated OCP concentrations in 60 fish samples from the brackish water of the Niger Delta. A total of six individuals of each of the following ten species: *Drapane africana*, *Mochokus niloticus*, *Chrysichthys nigrodigitatus*, *Pristipoma jubelini*, *Vome septapinis*, *Pseudotolithus senegalensis*, *Mugil cephalus*, *Pseudotolithus elongatus*, *Sphyraena piscatorum*, *and Lutjanus goreensis*, were investigated. The OCPs that were analysed were: α-BHC, β-BHC, γ-BHC, δ-BHC, endrin, endrin aldehyde, endrin ketone, heptachlor, heptachlor epoxide, aldrin, dieldrin, endosulfan I, endosulfan II, endosulfan sulfate methoxychlor, α-chlordane, γ-chlordane, DDE, DDT, and DDT. *Drapane africana* had the highest mean concentration of OCPs (∑4302 μg/kg fresh wt.), with a range of 2237–6368 μg/kg. The lowest concentration was found in *Mochokus niloticus* with a mean value of 2320 μg/kg and range of 1006–3288 μg/kg. The authors noted that the WHO/FAO guideline of 2000 μg/kg fresh weight was exceeded, suggesting a potential health risk to humans. The ten fish species were also analyzed for National Oceanic and Atmospheric Administration Agency (ΣNOAA) PCBs (sum of 27 congeners). The highest concentration of ΣNOAA PCBs (1830.0 ± 484.0 μg/kg) was detected in *Vomer septapinis*, while the lowest concentration (795 ± 169.3 μg/kg) was found in *Pseudotolithus senegalensis* [74].

These findings show that fish samples, irrespective of the water source (fresh or brackish water), contain significant levels of organochlorine pesticides. However, there is limited data available on levels of certain POPs of relevance to food safety in fish from Nigeria, particularly dioxins, brominated flame retardants, and fluorinated compounds.

##### Marine Water

POP levels in marine fish were reported by Osibanjo et al., (1990) [61], who analyzed 94 samples of 25 marine fish species between 1983 and 1985 and subsequently 14 samples of 7 shellfish species in 1987. Their study did not indicate any significant variation between the years but revealed higher concentrations of aldrin, heptachlor, HCB, and lindane in fish than in shellfish, while the levels of DDT and PCBs were higher in shellfish. They also showed that predatory fish had higher concentrations of pesticide residues in their muscle tissues than plankton feeders. Further studies on POP levels in fish and shellfish from Nigerian waters are warranted

### 3.3. Metals

Anthropogenic sources of heavy metals in aquatic ecosystems in Nigeria include effluents from the petroleum industry and agricultural discharge [75,76,77,78,79,80,81,82,83,84,85]. In some cases, the concentrations of metals were found to exceed the maximum permitted levels, implying potential health risks to aquatic organisms and human consumers. Nsofor et al. (2014) [40] investigated metal levels in the water and in catfish samples from three stations in the River Niger (Table 4). While the concentrations of zinc and copper in the fish were below the maximum limits [86], the iron concentrations in fish exceeded the limits in both the dry and wet seasons. Higher metal concentrations have been reported in both water samples and fish samples in the wet season compared to the dry season [87]. Similarly, the presence of Cd, Mn, Cr, Ni, Cu, Zn, and Pb have been reported in *Tilapia zillii* fish from the freshwater catchment area of the River Niger around the Ajaokuta Steel Company [88].

Olusola and Festus (2015) [42] assessed Cd, Cu, Cr, Ni, Pb, and Zn contaminants in the muscle, gills, eye, bone, and head of five fish species (*Pentanemus quinquarius*, *Pseudoltolithus senegalensis*, *Trichirus lepturus*, *Plectorhynchus meditarraneus*, and *Pseudotolithus typus*) from the coastal waters of Ondo State in the Ilaje local government area (LGA) along the Atlantic Ocean shoreline [42]. Cd (ranged from below the limit of detection (LOD) to 1.14 mg/kg)) and Pb (from <LOD to 0.71 mg/kg) were found to exceed the Codex Alimentarius maximum levels [86]. The coastal waters of the Ilaje LGA have been exposed to various oil exploration activities and industrial effluent, which may have impacted the aquatic organisms, therefore acting as a potential health hazard. Furthermore, the Ondo State coastal land area is rich in bitumen, and the high concentration of heavy metals and possibly PAHs in this region have been largely attributed to bitumen deposits [89].

Similarly, Abarshi et al., (2017) [90] assessed the presence of Cu, Ni, Zn, Pb, Mn, Fe, and Cd in the organs (liver, gills, and muscle) of fish samples obtained from the Finima Creek and Bonny River in Rivers State. High concentrations of heavy metals were found in fish samples from the two areas, while the highest concentrations were in the fish from the Finima Creek. Metal concentrations (µg/g dry wt.) in the fish muscle from the Finima Creek were Cu (5.75 ± 1.65); Zn (124.50 ± 4.34), Fe (565.60 ± 11.89), Mn (43.72 ± 3.42), Ni (30.00 ± 2.27), and Pb (5.00 ± 0.62), and the metal concentrations from the fish in the Bonny River were Cu (3.50 ± 0.77), Zn (14.00 ± 1.55), Fe (102.00 ± 4.20), Mn (9.34 ± 1.27), Ni (5.33 ± 0.96), and Pb (0.20 ± 0.05), respectively. The concentrations of Cu, Zn, and Fe were in the order of liver > gills > muscle, whereas Mn, Ni, Pb, and Cd distributions were in the order of gills > liver > muscle. These concentrations, except for Cd, exceeded the maximum levels in fish [86]. While Cd was not detectable, Pb concentrations in the samples from the Finima Creek were 100 times higher than the maximum level and four times higher than those in fish from the Bonny River. The authors attributed the high metal concentrations to the frequent oil spills and industrial effluents discharged into these rivers and the Finina River [90]. These creeks are known for pollution from oil spills and various forms of anthropogenic activities. The findings of Abarshi et al. (2017) [90] are comparable to those of Türkmena et al. (2009) [91]. These researchers reported that the metal contaminants found in the muscle of fish samples from the Aegean in the Mediterranean Sea, were generally lower than those obtained from the liver.

Arsenic was detected in four demersal fish species (*Chrysichthys nigrodigitatus*, *Mugil cephalus*, *Liza falcipinnis*, and *Bathygobious soporator)* from the Lagos Lagoon, with a positive correlation between the arsenic concentrations in fish muscle and water during the dry and wet seasons [44]. However, the health risk associated with the fish species is not considered significant since the target hazard quotient (THQ) was below 1, indicating that there is no cancer risk associated with the consumption of these fish species [92]. It was also noted that the level of arsenic found in the water during the wet and dry season did not exceed the (WHO) limit of 10 μg/L [86,92].

Concentrations of metals (Cd, Zn, Pb, and Hg) were analyzed in the gills, muscle and intestine from three fish species, *Tilapia zillii*, *Malapterurus electricus* and *Clarias gariepinus* from the River Niger in Onitsha, Anambra State [46]. The concentrations detected were far below the maximum levels set by the Codex Alimentarus [86]. Similarly, in their study of metals in *C. anguillaris*, *H. niloticus*, *and T. zilli* from Geriyo lake Yola Nigeria, Bawuro et al. [93] showed that the gills are the target organ for Zn, Cu, and Pb. On the other hand, Kamaruzzaman et al. (2010) [94] reported that the concentrations of Zn, Cu, and Pb follow the order stomach > muscle > gills in *S. leptolepis*, *E. affinis*, *P. niger*, *L. malabaricus*, *E. sexfasciatus*, *R. kanagurta*, *N. japonicus*, *and M. cordylafrom* from the Pahang coastal water in Malaysia. The concentrations of metals in several fish samples were found to be low [46,95,96,97,98]. Copat et al. (2013) [99] reported similarly low levels of Cd and Pb in shellfish samples from the eastern Mediterranean Sea. The maximum levels set by the European Commission for human consumption were not exceeded in the fish species that were analyzed. The liver and gills generally contain higher levels of metals compared to other organs as a result of environmental pollution. Generally, heavy metals are accumulated and biomagnified through the food chain. In aquatic systems, predatory fish generally have the highest levels of heavy metals, particularly mercury [100]. Apart from the human health risk associated with metal exposure from fish consumption, the potential effects on fish health warrant investigation, particularly in locations where there are large deposits of metals.

### 3.4. Microplastics

Over the last decade, marine plastic debris has become a global concern. The use of plastics dates back to the 20th century when it was first synthesized as a product called ‘Bakelite’, and its production began in earnest at the end of World War II and increased to about 5 million tons annually [101,102]. Over the years, waste generated from plastics has become a major environmental concern, particularly in marine ecosystems [103]. The biodegradation of plastics is slow, particularly in the ocean, where the temperature is low [104].

#### Microplastics in Fish

The annual input of plastic entering the ocean from waste generated by 192 coastal countries worldwide has been quantified [105], and Nigeria was ranked 9th, producing 13% of global plastic waste, 0.85 MMT/year of mismanaged plastic waste (2.7% of the global total) and 0.13–0.34 MMT/year of plastic marine debris. Nigeria reportedly generated 42 million tons of solid waste of the total 62 million tons generated by Sub-Saharan Africa [106,107]. About 20% of the landfills in Nigeria is waste from polythene materials, plastic water bottles, water sachets, and plastic bags [108,109,110]. Lagos alone contributes 450,000 tons of plastic waste that enter into the ocean environment annually [111].

There is very limited data available regarding microplastics in fish from Nigeria; however, Adeogun et al. (2020) [48] recently published the first report on microplastics in the stomachs of commonly consumed fish species from a municipal water source (Eleyele Lake) in Ibadan (Oyo State) in southwestern Nigeria. They highlighted the presence of ingested plastics of various sizes in the fish stomachs. A total of 109 fish samples belonging to eight species (*Coptodon zillii*, *Oreochromis niloticus*, *Sarotheron melanotheron*, *Chrysicthys nigrodigitatus*, *Lates niloticus*, *Paranchanna obscura*, *Hemichromis fasiatus*, and *Hepsetus odoe*) were analyzed from different habitats and trophic levels. All of the species except *Hemichromis fasiatus* had microplastics in their stomachs. The highest occurrence of microplastics was in the benthopelagic fish species, *O. niloticus* (34%), *C. zillii* (32%), and *S. melanotheron* (13%). The lowest occurrence was found in *P. obscura*, *L. niloticus*, and *H. odoe* (5%). The sizes of the microplastic particles detected in the fish ranged from 1 µm–1.5 mm. The largest range in particle size (126 µm–1.5 mm) was detected in *C. zillii*, and the lowest (1–1.53 µm) was detected in *H. odoe*. Feeding on plankton and aquatic plants were correlated with the wide range of microplastic sizes. It was also shown that feeding mode and trophic levels were important variables for the particle size of the ingested microplastics. A higher occurrence of microplastics was found in the benthopelagic species compared to pelagic/demersal species [55], which is comparable to findings of other studies [112,113,114,115,116].

Similar findings were reported from Ghana by Adika et al. (2020) [117]. These researchers showed that industrially produced pellets were the most prevalent microplastics in the fish species that were analyzed (31%) followed by microbeads (29%) and burnt film plastics (22%) while microfibers (2%), threads (2%), and foams (<0.1%) were the least frequently occurring microplastic particles.

Even though ingested plastics have been reported in the stomach contents of fish from Nigeria, their presence in muscle tissue and their potential impact on food safety remains unclear. Other potential health challenges of microplastics in seafood are with associated chemicals such as bisphenol A, phthalates, flame retardants, and other toxic monomers [118,119] as well as hydrophobic contaminants (POPs) and pathogens [120,121,122,123,124,125].

Other challenges include:The ability to establish the mechanisms by which bioaccumulated plastics through the food chain could affect humans;An evidence-based approach in determining the levels of chemicals from plastic-waste that are detrimental to humans;The identification of humans who are more vulnerable to the impacts of plastic waste;How to balance the cost and benefits of mitigating these problems [123,126].

In response to some of these challenges, Nigerian lawmakers passed a bill on the use of plastic bags called the “Plastic Bags (Prohibition) Bill 2018”, where action was to be taken to reduce pollution caused by discarded polythene bags on landfills, thereby protecting both people and the environment. However, there have been no sustainable strategies for the effective implementation of the bill, coupled with the weak structure of the regulatory agencies for the control of waste generated from plastics.

## 4. Conclusions

This report includes the contaminant levels of 49 different species of fish from coastal Nigeria even though Nigeria has more than 104 different species of commercial importance, including inland and farmed fish [8], demonstrating the lack of published data on many species. The main locations associated with pollutants are the coastal waters of the Lagos Lagoon in Lagos State (which serves as a sink for most of the industrial effluent and domestic water waste), the coastal area of the Ogun and Ondo States, and the South-South region. In general, heavy metal levels in fish are below maximum levels, while there is limited data available on POPs of relevance to food safety in fish from Nigeria, particularly dioxins, brominated flame retardants, and fluorinated compounds.

There are challenges highlighted in this report that must be considered with regard to contaminants in fish. The bill on the use of plastic bags, the “Plastic Bags (Prohibition) Bill 2018”, may have to be revisited with a better practical, effective, and measurable implementation strategy. While the reduction of anthropogenic activities may have an impact on the discharge of plastics to the aquatic environment, the government should promote the production and use of alternatives to plastic materials (e.g., glass and biodegradable materials). Government intervention for the effective management or control of pollutants in the environment by different regulatory agencies should be strengthened. Given the high demand and consumption of fish in Nigeria, there is a need to guarantee its safety and quality. This review provides valuable information on the levels of chemical contaminants in fish from the coastal area of Nigeria and serves as a baseline for further research on contaminants in fish from the West African coast.

## Figures and Tables

**Figure 1 foods-10-02013-f001:**
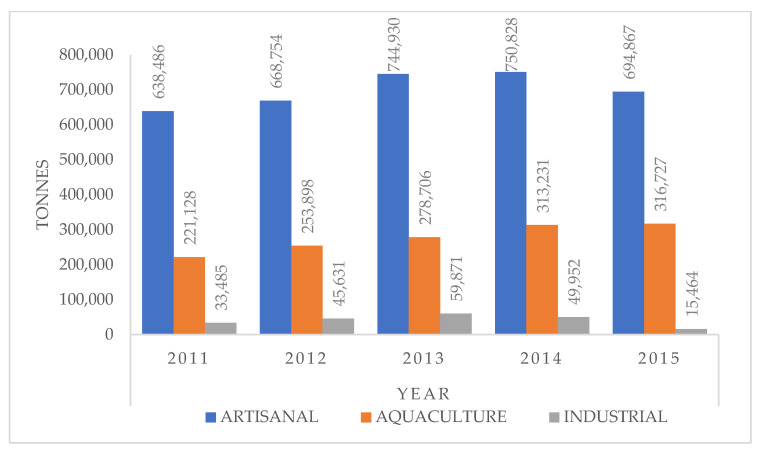
Fish production (metric tons) in Nigeria for 2010–2015. (Adapted from National Bureau of Statistics, 2017 [8]).

**Table 1 foods-10-02013-t001:** Commercially important fish families in Nigeria.

Environment	Common Name	Family	Species
Marine fish	Croaker	Sciaenidae	*Pseudotolithus typus*, *Pseudotolithus senegalensis*, *Pseudotolithus elongatus*, *Pseudotolithus senegalensis*, *Brachydeuterus auratus*, *Selene setapinnis*
	Catfish	Ariidae	*Carlarius heudelotii* *Arius gigas*, *Arius latiscutatus*, *Arius parkii*
	Grunters	Haemulidae	*Pomadasys jubelini*, *Pomadasys suillus* *Pomadasys incisus*, *Pomadasys perotaei*
	Tongue Sole	Cynoglossidae	*Cynoglossus senegalensis*, *Cynoglossus canariensis* *Cynoglossus monodi*, *Cynoglossus browni*
	Threadfins	Polynemidae	*Polydactylus quadrifilis* *Galeoides decadactylus*
	Jackfish	Carangidae	*Caranx hippos*, *Caranx crysos* *Caranx latus* *Caranx lugubris*
	Barracudas	Sphyraenidae	*Sphyraena barracuda*, *Sphyraena afra*, *Sphyraena guachancho*
		Clupeidae	*Sardinella* spp.
	Red snappers	Lutjanidae	*Lutjanus goreensis* *Lutjanus fulgens* *Lutjanus agennes* *Lutjanus dentatus*
	Groupers	Serranidae	*Epinephelus aeneus*
		Sparidae	*Dentex canariensis* *Dentex angolensis* *Dentex congoensis*
	Breams		*Pagrus* spp., *Pagellus bellottii*, *Pagus* spp.
Fresh water fish		Mormyridae	43 spp.
		Mochokidae	27 spp.
		Characidae	25 spp.
		Cichlidae	19 spp.
		Cyprinidae	32 spp.

Source: Olaoye and Ojebiyi [4].

**Table 2 foods-10-02013-t002:** Fish production (metric tons) from different sectors.

S/No		SECTOR/YEAR	2011	2012	2013	2014	2015
1	ARTISANAL	Coastal and Brackish water	346,381	370,918	418,537	435,384	382,964
		Inland: Rivers and Lakes	292,105	297,836	326,393	324,444	311,903
		Sub-Total	638,486	668,754	744,930	759,828	694,867
2	AQUACULTURE	Sub-Total	221,128	253,898	278,706	313,231	316,727
3	INDUSTRIAL	Fish (Inshore)	19,736	27,977	37,652	29,237	10,727
		Shrimp (Inshore)	13,749	17,654	22,219	20,715	4737
		EEZ	-	-	-	-	-
		Sub-Total	33,485	45,631	59,871	49,952	15,464
	TOTAL		893,099	968,283	1,083,507	1,123,011	1,027,058

Source: Nigeria’s Fish Production: 2010–2015 [8].

**Table 3 foods-10-02013-t003:** Summary of the fish species and their sources assessed for contaminants.

Fish Species	Sample Size	Location	Analytical Method Used	Reference	Year Published
PAHs					
*Periophthalmus* *koelreuteri*	* 30	Akwa-Ibom State	GC/MS, GC/FID	[17] [18]	2013 2016
*Caranx hippos* *Chrysichthys nigrodigitatus* *Lutjanus dentatus* *Ethmalosa fimbriata* *Lutjanus agennes* *Mugil cephalus* *Pomadasys jubelini* *Sarotherodon melanotheron* *Scomberomorus trito* *Sphyraena barracuda* *Tarpon atlanticus* *Tilapia guineensis*	12	Lagos Lagoon	GC/MSD NA	[19] [20]	2012 2015
*Arius heudeloti* *Cynoglossus senegalensis* Haake	-	Lagos State (smoked fish)	GC/FID HPLC GC/FID	[18] [21] [22]	2016 2011 2021
*Arius heudeloti* *Blunt hwake* *Clarias gariepinus* *Cynoglussus s enegalensis* *Mud minnow*	-	Benue State	GC/MS NA	[23] [24]	2015 2013
*Clarias gariepinus* *Ethmalosa fimbriata* *Scomber scombrus* *Tilapia zilli*	-	Edo State	GC/MSD GC/FID	[19] [25]	2012, 2017
POPs					
Freshwater fish samples (Species not classified)	40	Oyo State (Freshwater sources)	NA GC/MS	[26] [27]	2019 2008
*Oreochromis niloticus* *Sarotherodon galileaus* *Tilaipia guineensis*,	1074	Ogun River, Ogun State	HPLC GC/FID	[21] [28]	2011 2019
*Clarias gariepinus* *Tilapia zilli*	92	Edo State	GC/ECD GC	[29] [30]	2015 2019
*Clarias gariepinus* *Neochanna diversus*	92	Edo State	GC/ECD, GC/MS-MS	[31] [32]	2009 2019
*Sarotherodon melanotheron*	155	Lagos Lagoon	GC/MS NA	[33] [34]	2019 2005
*Chrysichthys nigrodigitatus* *Ethmalosa fimbriata* *Tilapia zilli*	-	Lagos Lagoon	GC/ECD NA	[35] [36]	2008 2001
*Pseudotolithus semegalensis Pseudotolithus typus*	-	Lagos Lagoon	GC/ECD GC	[37] [38]	2013 1997
*Lutjanus goreensis* *Regalecus glesne* *Sardinella maderensis*	-	Lagos lagoon	GC/MS GC/ECD	[23] [39]	2015 2018
*Chrysichthys nigrodigitatus Pristipoma jubelini* *Drapane Africana* *Lutjanus goreensis* *Mochokus niloticu* *Mugil cephalus* *Pseudotolithus elongatus* *Pseudotolithus senegalensi* *Sphyraena piscatorum* *Vome septapinis*	60	Niger Delta	GC/FID GC/ECD	[25] [39]	2017 2018
METAL CONTAMINANTS					
*Clarias gariepinus*	-	River Niger	AAS NA	[40] [41]	2014 2001
*Plectorhynchus meditarraneus Pseudotolithus typus* *Pentanemus quinquarius Pseudoltolithus senegalensis Trichirus lepturus*	-	Ondo State	AAS NA	[42] [43]	2015 2020
*Bathygobious soporator* *Chrysichthys nigrodigitatus* *Liza falcipinnis* *Mugil cephalus*	-	Lagos Lagoon	ICP-MS NA	[44] [45]	2017 2014
*Clarias gariepinus* *Malapterurus electricus* *Tilapia zillii*	-	Anambra State	AAS NA	[46] [47]	2019 2001
MICROPLASTICS					
*Coptodon zillii*, *Hemichromis fasiatus*, *Hepsetus odoe* *Lates niloticus* *Oreochromis niloticus* *Paranchanna obscura* *Sarotheron melanotheron Chrysicthys nigrodigitatus*	109	Oyo State (municipal water)	FSCM NA	[48] [49]	2020

* The samples also included other types of seafood (*Crassostrea virginica* and *Littorina littorea).* NA: Not Applicable (the publication is not a technical paper).

**Table 4 foods-10-02013-t004:** Concentrations of zinc (Zn), iron (Fe), copper (Cu), and lead (Pb) in river water and catfish from the River Niger.

Metal	Dry Season		Wet Season		WHO (1984)
	River Niger	Catfish	River Niger	Catfish	
Zn	0.261 ± 0.066	3.5 ± 0.432	0.36 ± 0.089	4.423 ± 0.693	5
Fe	1.41 ± 1.182	4.73 ± 0.221	1.59 ± 1.306	5.731± 1.205	0.3
Cu	0.008 ± 0.005	0.60 ± 0.113	0.05 ± 0.077	1.14 6 ± 0.343	1
Pb	ND	0.04 ± 0.007	ND	0.416 ± 0.472	0.05

Source: Nsofor and Ikpeze (2014) [40].

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
