# Peer review of "A Review of Chemical Contaminants in Marine and Fresh Water Fish in Nigeria"

_foods, 2021, doi:10.3390/foods10092013_

Round 1

Reviewer 1 Report

The MS introduces a comprehensive review on the trace metal, microplastics and POPs contents of the principal fish species commercialized and consumed in Nigeria. The paper appears limited and attributable to a simple regional study.

However, the paper could give valuable information on the risks related to the consumption of over 40 fish species, giving detailed results. All my suggestions are listed below:

Abstract

The abstract need to be improved. First of all, AA should emphasize the scope of the paper by highlighting the actual critical issues on this topic. Furthermore, AA should report what are the fish species which revealed the highest concentrations of contaminants sorting by their ecology and habitat distribution.

Lines 21-22: Superfluous information, delete this sentence.

Introduction

The introduction takes too long in the geographical description of Nigeria (e. g. lines 32-35, lines 37-44, lines 45-48), I suggest to go straightforward to the core of the study.

Figure 1 is superfluous, I suggest to delete it.

Even the lines 62-65 are out of the scope of the paper. Furthermore, I would not consider the division in artisanal and industrial fish (lines 71-75).

Materials and Methods

AA must report the time covering of the studies considered.

Lines 112-113: I retain that the chronological information is always relevant to understand the trend of the pollutants under study.

Results and Discussion

Lines 123-126: This part should be added in the introduction section.

It is important that AA report every method used by the studies considered in order to have a reliable comparison.

AA cited European legislation for PAH and metals limits; given this I think it could be interesting to compare the results of Nigerian studies with what was found in European farmed and wild fish. I suggest https://doi.org/10.1016/j.foodchem.2008.06.071; https://doi.org/10.1016/j.fct.2012.11.038; https://doi.org/10.3390/ani9090594; https://doi.org/10.1016/j.scitotenv.2012.11.082.

Lines 375-377: This part appear so confusing Cd was not mentioned before.

Lines 387-388: The references appear incomplete; being an index indicated by EFSA  I suggest to cite https://doi.org/10.1006/eesa.1996.0095;  https://doi.org/10.3389/fmars.2021.616488.   

Lines 395-399: The comparison is inappropriate because it refers to different fish species

Lines 400-401: “The liver and gills generally contain higher levels of metals compared to other organs, as a results of environmental pollution” …This sentence does not properly justify the difference in the accumulation of trace metals in the different organs. The authors should take into greater consideration the physiology and the roles of each organ within fish homeostasis to better understand the differences in the accumulation of heavy metals without however excluding the chemical nature of each of them.

Lines 402-403: AA should be more precise regarding this aspect. First of all AA should pay attention on the biomagnification phenomena in marine environment https://doi.org/10.3390/ani10091663; https://doi.org/10.3390/toxins11110620. The effects of biomagnification can be seen in top predators, not in small pelagic fishes.

Lines 408-419: These information are superfluous for the scope of the paper.

Line 455: AA should take into account the relation between the size of the organisms and the presence of contaminants. I suggest https://doi.org/10.1016/j.scitotenv.2019.04.196

Author Response

Dear Editor,

Thank you for letting us submit a revised version of our manuscript, and we appreciate the valuable and constructive comments from the reviewers. The manuscript has been updated according to the reviewers’ suggestions, and the responses are listed below.

Reviewer 1

The MS introduces a comprehensive review on the trace metal, microplastics and POPs contents of the principal fish species commercialized and consumed in Nigeria. The paper appears limited and attributable to a simple regional study.

However, the paper could give valuable information on the risks related to the consumption of over 40 fish species, giving detailed results. All my suggestions are listed below:

Abstract

The abstract need to be improved. First of all, AA should emphasize the scope of the paper by highlighting the actual critical issues on this topic. Furthermore, AA should report what are the fish species which revealed the highest concentrations of contaminants sorting by their ecology and habitat distribution. We have revised the abstract and hope that both scope and main findings are clearer in the present version.

Lines 21-22: Superfluous information, delete this sentence. Sentence is deleted.

Introduction

The introduction takes too long in the geographical description of Nigeria (e. g. lines 32-35, lines 37-44, lines 45-48), I suggest to go straightforward to the core of the study. The introduction addresses the fish production status in Nigeria as a general background information, which is relevant while discussing contaminants. The paragraph has been shorted as suggested, but the references are retained.

If Lines 123-126 on PAHs are to be inserted here as suggested, then background information on other contaminants should also be part of this section. I think this may be too long for an Introduction.

Figure 1 is superfluous, I suggest to delete it. Figure 1 has been deleted.

Even the lines 62-65 are out of the scope of the paper. Furthermore, I would not consider the division in artisanal and industrial fish (lines 71-75). Information in line 62-65 is considered relevant, because it is along the creeks and lagoons most of the fish contaminants  are discharged. It also gives information  on the artisanal fishery as relevant source of  contaminated fish.  It is important to have these statistics, it can be related to fish the demand.

Materials and Methods

AA must report the time covering of the studies considered.

There is no restriction as per the timeline for this review. However data giving current information were preferred.  

Lines 112-113: I retain that the chronological information is always relevant to understand the trend of the pollutants under study.

Results and Discussion

Lines 123-126: This part should be added in the introduction section. Lines 123-126 have been deleted, but not included in the introduction as suggested. Please see our comment under the introduction section.

It is important that AA report every method used by the studies considered in order to have a reliable comparison.

We agree in that the analytical method used may influence the results. However, there are several factors, e.g. sampling schemes, tissue analysed, handling and storage of samples pending analyses, sample size to mention some, that all together have impact on the quality of the data reported. Thus, method used is one of several factors that may affect reliable comparison.

We have acknowledged this in the Materials and methods section of the paper, lines: 127-129.

AA cited European legislation for PAH and metals limits; given this I think it could be interesting to compare the results of Nigerian studies with what was found in European farmed and wild fish. I suggest https://doi.org/10.1016/j.foodchem.2008.06.071; https://doi.org/10.1016/j.fct.2012.11.038; https://doi.org/10.3390/ani9090594; https://doi.org/10.1016/j.scitotenv.2012.11.082. We think this is outside the scope of this work. Nigeria and European farmed and wild fish are not within the same ecological or geographical zone, furthermore, the fish species, and the environmental hazards are different. Even within Nigeria contaminants in  farmed and wild fish in Lagos  state is different from that in the Delta region and that from the Cross-river state. Comparison in a study like this is usually to the global standards and not between particular regions.

Lines 375-377: This part appear so confusing Cd was not mentioned before. This was an oversight, Cd has been included (line 368).

Lines 387-388: The references appear incomplete; being an index indicated by EFSA  I suggest to cite https://doi.org/10.1006/eesa.1996.0095;  https://doi.org/10.3389/fmars.2021.616488. It is unclear what the reviewer means, can you please clarify; EFSA does not appear as a reference here and the suggested references do not appear to fit with the text in the manuscript.

Lines 395-399: The comparison is inappropriate because it refers to different fish species. The comparison here refers to the fish body parts that are more susceptible to harboring contaminants, irrespective of the type of fish. There are several reports substantiating that the gills and the liver have highest concentrated of metal contaminants, irrespective of fish type.

Lines 400-401: “The liver and gills generally contain higher levels of metals compared to other organs, as a results of environmental pollution” …This sentence does not properly justify the difference in the accumulation of trace metals in the different organs. The authors should take into greater consideration the physiology and the roles of each organ within fish homeostasis to better understand the differences in the accumulation of heavy metals without however excluding the chemical nature of each of them. Information based on the article reviewed and properly cited showed that “the liver and gills generally contain higher levels of metals compared to other organs, as a result of environmental pollution”. The authors of the different articles reviewed have fully discussed and  justified the differences in the accumulation of metals in the organs. Other factors may prevail,

Lines 402-403: AA should be more precise regarding this aspect. First of all AA should pay attention on the biomagnification phenomena in marine environment https://doi.org/10.3390/ani10091663; https://doi.org/10.3390/toxins11110620. The effects of biomagnification can be seen in top predators, not in small pelagic fishes.

We agree in that the reviewer raise an important issue. However,  https://doi.org/10.3390/ani10091663 refers to contaminants associated with rare earth elements containing phosphate fertilizers, mining, and dispersion from indigenous rocks, while https://doi.org/10.3390/toxins11110620 refers to environmental factors leading to the formation of different microcystins congeners in freshwaters. Presently, there are no published  studies in this area in Nigeria. However,  this present work referred to  runoffs  of pesticides from farmlands into rivers and lakes where the bioaccumulation in fish occurred within the inland waters and not marine . This was mentioned in lines 253-254 and 263-267. The issue of bioaccumulation or biomagnification from some other sources were well referenced  [see Ref 25, 53, 98 and 100]. 

Lines 408-419: This information are superfluous for the scope of the paper. This is a background information without which, no one will worry about the menace of plastics in the environment. The second paragraph is removed, but we think some readers will appreciate some of this information to be able to comprehend the problems associated with plastics. Lines 415-420 have been deleted.

Line 455: AA should take into account the relation between the size of the organisms and the presence of contaminants. I suggest https://doi.org/10.1016/j.scitotenv.2019.04.196

Thanks for this suggestion. The reference is inserted.

Reviewer 2 Report

ARTICLE TITLE:  A Review of Chemical Contaminants and Microplastics in Marine and Fresh Water Fish in Nigeria

ORIGINALITY: The review can be considered as a contribution to knowledge on the concentrations of Polycyclic aromatic hydrocarbons (PAHs), persistent organic pollutants (POPs), metals, and microplastics in fish consumed in Nigeria.

SIGNIFICANCE: Significant

QUALITY OF PRESENTATION: The paper is well-written and the results are linked to the main theme attempting to summarize findings and the challenges highlighted in this review.

SCIENTIFIC SOUNDNESS: Diverse literature were used in the work. Attempts should be made to cite more current literature in the last 3-5 years.

INTEREST TO READERS: Appropriate.

OVERALL MERIT: Good

ENGLISH LEVEL: Needs improvement

OVERALL RECOMMENDATIONS: Accept subject to appropriate revisions.

SUMMARY

The paper is aimed at highlighting findings on contaminants in fish consumed in Nigeria. Of interest to the paper, are reports highlighting the levels of environmental pollution in freshwater and marine fish in Nigeria. The major contaminants reviewed in this paper are Polycyclic aromatic hydrocarbons (PAHs), persistent organic pollutants (POPs), metals, and microplastics.

SPECIFIC COMMENTS

  1. 1, line 18: Consider providing the full name of the PAHs, initials can follow subsequently.
  2. 2, lines 57: Please provide reference(s) for the statement.
  3. 8, lines 212-214: Please provide reference(s) for the statements.

Be consistent with your pattern of reference. i.e provide the missing DOI for some of the references

Author Response

REVIEW REPORT, Reviewer 2

ARTICLE TITLE:  A Review of Chemical Contaminants and Microplastics in Marine and Fresh Water Fish in Nigeria

ORIGINALITY: The review can be considered as a contribution to knowledge on the concentrations of Polycyclic aromatic hydrocarbons (PAHs), persistent organic pollutants (POPs), metals, and microplastics in fish consumed in Nigeria.

SIGNIFICANCE: Significant

QUALITY OF PRESENTATION: The paper is well-written and the results are linked to the main theme attempting to summarize findings and the challenges highlighted in this review.

SCIENTIFIC SOUNDNESS: Diverse literature were used in the work. Attempts should be made to cite more current literature in the last 3-5 years.

Thank you for this comment. We have now included three more references from recent years.

INTEREST TO READERS: Appropriate.

OVERALL MERIT: Good

ENGLISH LEVEL: Needs improvement

The language has been improved.

OVERALL RECOMMENDATIONS: Accept subject to appropriate revisions.

SUMMARY

The paper is aimed at highlighting findings on contaminants in fish consumed in Nigeria. Of interest to the paper, are reports highlighting the levels of environmental pollution in freshwater and marine fish in Nigeria. The major contaminants reviewed in this paper are Polycyclic aromatic hydrocarbons (PAHs), persistent organic pollutants (POPs), metals, and microplastics.

SPECIFIC COMMENTS

  1. 1, line 18: Consider providing the full name of the PAHs, initials can follow subsequently. Poly aromatic hydrocarbons has been written in full in line 18.
  2. 2, lines 57: Please provide reference(s) for the statement. The reference is as cited under Table 1. Olaoye and Ojebiyi [8]
  3. 8, lines 212-214: Please provide reference(s) for the statements. Reference is provided as, Miniero, R., Iamiceli, A. L. Persistent Organic Pollutants. In Encyclopedia of Ecology, Academic Press. 2008, pp 2672-2682. Available online https://doi.org/10.1016/B978-0-12-409548-9.09496-3

Be consistent with your pattern of reference. i.e provide the missing DOI for some of the references. Most of the DOI as given by the publishers and accessible have been provided.

Round 2

Reviewer 1 Report

AA have not replied satisfactorily to my suggestion; some examples are shown below:

1) There is no restriction as per the timeline for this review. However data giving current information were preferred. R: what does it means this? Most of the review reported in literature regarding this topic (contaminants in fish) highlight the temporal trend on the presence of contaminants (see https://doi.org/10.1016/j.chemosphere.2007.05.100; https://doi.org/10.1016/S0048-9697(99)00038-8 ; https://doi.org/10.1016/j.jglr.2009.12.008 ; https://doi.org/10.1021/es035288h ; https://doi.org/10.1016/j.scitotenv.2004.03.022 ) therefore, I retain that this aspect is essential for this type of paper; AA should add these information.

2) We agree in that the analytical method used may influence the results. However, there are several factors, e.g. sampling schemes, tissue analysed, handling and storage of samples pending analyses, sample size to mention some, that all together have impact on the quality of the data reported. Thus, method used is one of several factors that may affect reliable comparison. R: If AA consider this probably they should question the entire manuscript as this heterogeneity would not lead to an harmonized overview of the data consulted; therefore, I suggest to consider this suggestion.

3) We think this is outside the scope of this work. Nigeria and European farmed and wild fish are not within the same ecological or geographical zone, furthermore, the fish species, and the environmental hazards are different. Even within Nigeria contaminants in  farmed and wild fish in Lagos  state is different from that in the Delta region and that from the Cross-river state. Comparison in a study like this is usually to the global standards and not between particular regions. R: Since the AAs appeal to the ecosystemic diversity between the Mediterranean countries and Nigeria, why did they use parameters and limits dictated by the European Union that are in agreement with the productive realities and the ecological findings of European countries? They should refer to their legislation or else add a comparison with data from Europe

4) The effects of biomagnification can be seen in top predators, not in small pelagic fishes. R: So why the authors write "Generally larger, and longer lived species higher in the food chain have higher metal concentrations than smaller or shorter-lived species of fish"? AA should refer to this phenomena and give references on this.

Author Response

  • There is no restriction as per the timeline for this review. However data giving current information were preferred. R: what does it means this? Most of the review reported in literature regarding this topic (contaminants in fish) highlight the temporal trend on the presence of contaminants (see https://doi.org/10.1016/j.chemosphere.2007.05.100; https://doi.org/10.1016/S0048-9697(99)00038-8 ; https://doi.org/10.1016/j.jglr.2009.12.008 ; https://doi.org/10.1021/es035288h ; https://doi.org/10.1016/j.scitotenv.2004.03.022 ) therefore, I retain that this aspect is essential for this type of paper; AA should add these information.

Thank you for this comment. We agree in timelines are highly relevant and have now included year published in Table 3. We did not include any time restrictions in the literature search and added text (line 129-137). In order to make timelines, data is needed. In the case of Nigeria, we identified large data gaps and data scarcity. Thus, we conclude (abstract) “This work revealed the need for a more systematic sampling strategy for fish in order to identify the most vulnerable species, the hot spots of contaminants, and applicable food safety control measures for fish produced and consumed in Nigeria”.

  • We agree in that the analytical method used may influence the results. However, there are several factors, e.g. sampling schemes, tissue analysed, handling and storage of samples pending analyses, sample size to mention some, that all together have impact on the quality of the data reported. Thus, method used is one of several factors that may affect reliable comparison. R:If AA consider this probably they should question the entire manuscript as this heterogeneity would not lead to an harmonized overview of the data consulted; therefore, I suggest to consider this suggestion.

We agree, and have now included both method and year published in Table 3.

  • We think this is outside the scope of this work. Nigeria and European farmed and wild fish are not within the same ecological or geographical zone, furthermore, the fish species, and the environmental hazards are different. Even within Nigeria contaminants in  farmed and wild fish in Lagos  state is different from that in the Delta region and that from the Cross-river state. Comparison in a study like this is usually to the global standards and not between particular regions. R:Since the AAs appeal to the ecosystemic diversity between the Mediterranean countries and Nigeria, why did they use parameters and limits dictated by the European Union that are in agreement with the productive realities and the ecological findings of European countries? They should refer to their legislation or else add a comparison with data from Europe.

Thanks for this comment. The national legislations based on international standard, for the purposes of e.g. international trade. We have now compared some of the results with data from Europe and included new references.

4) The effects of biomagnification can be seen in top predators, not in small pelagic fishes. R: So why the authors write "Generally larger, and longer lived species higher in the food chain have higher metal concentrations than smaller or shorter-lived species of fish"? AA should refer to this phenomena and give references on this.

Thanks for this comment. We have now acknowledged in the text that biomagnification can be seen in top predators (line 440, reference 108).